# Revisiting Multimodal Positional Encoding in Vision–Language Models

**Jie Huang**[1,2,*,§]    **Xuejing Liu**[1,*]    **Sibo Song**[1]    **Ruibing Hou**[2,†]
**Hong Chang**[2]    **Junyang Lin**[1]    **Shuai Bai**[1,†]

[1]Qwen Team 🦀, Alibaba Group.
[2]Institute of Computing Technology, Chinese Academy of Sciences
[*]Equal contribution    [†]Corresponding author
{yuzheng.lxj,sibo.ssb,junyang.ljy,baishuai.bs}@alibaba-inc.com
{huangjie24s,houruibing,changhong}@ict.ac.cn

## Abstract

Multimodal position encoding is essential for vision-language models, yet there has been little systematic investigation into multimodal position encoding. We conduct a comprehensive analysis of multimodal Rotary Positional Embedding (RoPE) by examining its two core components: position design and frequency allocation. Through extensive experiments, we identify three key guidelines: positional coherence, full frequency utilization, and preservation of textual priors—ensuring unambiguous layout, rich representation, and faithful transfer from the pre-trained LLM. Based on these insights, we propose Multi-Head RoPE (MHRoPE) and MRoPE-Interleave (MRoPE-I), two simple and plug-and-play variants that require no architectural changes. Our methods consistently outperform existing approaches across diverse benchmarks, with significant improvements in both general and fine-grained multimodal understanding. Code is avaliable at https://github.com/JJJYmmm/Multimodal-RoPEs.

## 1 Introduction

The permutation-invariant nature of the self-attention mechanism requires the use of positional encodings to inform Large Language Models (LLMs) of sequence order, relative distance, and structural dependencies. While early methods relied on absolute position embeddings (Vaswani et al., 2017), relative encodings—which better generalize to varying sequence lengths—have become the standard. Among these, Rotary Position Embedding (RoPE) (Su et al., 2024) has emerged as the de facto choice in modern LLMs such as Llama (Grattafiori et al., 2024) and Qwen (Yang et al., 2025).

Vision-Language Models (VLMs) also require positional encodings that can handle heterogeneous modalities, including 1D text and 2D/3D visual inputs. Current methods fall into two main categories: 1D sequential and multi-dimensional designs. The former, exemplified by vanilla RoPE (Su et al., 2024) and V2PE (Ge et al., 2024), flattens and concatenates all inputs into a single sequence. While simple, this approach discards the native visual geometry, leading to a significant degradation in performance on tasks requiring visual grounding and spatial reasoning.

Multi-dimensional designs, the second approach, extend RoPE to multiple axes (time, height, width) by partitioning embedding channels. Qwen2-VL Wang et al. (2024a) adopts Multimodal RoPE (MRoPE) to unify positional encoding for text and visual tokens. However, MRoPE allocates the position embedding into t-h-w chunk, placing the temporal infomation entirely in the high-frequency channels. This bias in temporal encoding harms long-range video modeling. Subsequent work has attempted to improve it, but this has led to a fragmented landscape of highly specialized solutions. Some methods focus exclusively on image understanding (Wang et al., 2025), others on video comprehension (Wei et al., 2025; Li et al., 2025; Liu et al., 2025), and a third group on image generation (Liao et al., 2025; Wu et al., 2025). While these models achieve notable performance in their

---

[§]This work was done during a research internship at Qwen Team, Alibaba Group.

Table 1: Comparison of different RoPE methods.

| Method | Position Design | | Freq Allocation | | Compatible with Text-only RoPE |
|---|---|---|---|---|---|
| | 3D Struct. | Modal Int. | Range | Gran. | |
| Vanilla RoPE (Su et al., 2024) | ✗ | ✓ | - | - | ✓ |
| V2PE (Ge et al., 2024) | ✗ | ✓ | - | - | ✓ |
| RoPE-tie (Su, 2024) | ✓ | ✗ | ✓ | ✗ | ✓ |
| MRoPE (Bai et al., 2025) | ✓ | ✓ | ✗ | ✗ | ✓ |
| CircleRoPE (Wang et al., 2025) | ✓ | ✗ | ✗ | ✗ | ✓ |
| VideoRoPE (Wei et al., 2025) | ✓ | ✗ | ✗ | ✗ | ✓ |
| IL-RoPE (Liao et al., 2025) | ✓ | ✓ | ✗ | ✗ | ✗ |
| Omni-RoPE (Wu et al., 2025) | ✓ | ✓ | ✗ | ✗ | ✗ |
| MHRoPE | ✓ | ✓ | ✓ | ✓ | ✓ |
| MRoPE-I | ✓ | ✓ | ✓ | ✗ | ✓ |

respective domains, the development of a truly robust and versatile VLM requires a more holistic positional encoding strategy. In this work, we aim to develop a more holistic positional encoding strategy capable of supporting the core, unified capabilities of image and video understanding, complemented by fine-grained visual grounding.

To build a more robust multimodal positional encoding, we build on MRoPE and systematically explore three underexplored design: (i) position design—how to assign unambiguous, well-separated coordinates to text and visual tokens; and (ii) frequency allocation—how to distribute rotary frequencies across embedding dimensions for each positional axis; (iii) compatibility with text-only RoPE—ensuring the design defaults to vanilla RoPE for pure text inputs to enable effective transfer learning. As Table 1 shows, we systematically compare recent methods across the three design axes and conduct extensive experiments. From this analysis, we identify common pitfalls: modalities confusion arising from positional ambiguity; degraded cross-modal fusion due to suboptimal modality intervals; impaired multi-scale modeling from restricted frequency allocations; and compromised transfer learning caused by incompatibility with text-only RoPE.

Based on our experiment, we distill three core guidelines for designing robust VLM positional encodings: (i) positional coherence, requiring unambiguous coordinates with a well-defined modality interval; (ii) full frequency allocation, ensuring all positional axes have access to the full frequency spectrum; and (iii) preservation of textual priors, keeping the text RoPE identical to the base LLM. To satisfy the guidelines of full frequency allocation, we propose two simple yet effective methods. Multi-Head RoPE dedicates distinct attention heads to different positional axes to preserve full frequency resolution. MRoPE-Interleave employs a fine-grained, round-robin distribution of channels to ensure each axis is encoded with the full frequency spectrum. Besides, we introduce *spatial-reset*, a novel mechanism that resets the spatial position for visual content. This simple modification was found to significantly facilitate the model's focus to visual information.

Our methods consistently outperform strong baselines across key tasks, including image and video understanding and visual grounding. Our contributions are three-fold: (1) a systematic decomposition of multimodal RoPE design; (2) two lightweight instantiations—Multi-Head RoPE and MRoPE-Interleave that satisfy the guidelines; and (3) *spatial-reset*, a general-purpose optimization for improved visual information flow.

## 2 ANALYSIS OF MULTIMODAL ROTARY POSITION EMBEDDING

This section provides a systematic analysis of multimodal RoPE. We begin by revisiting the basics of vanilla RoPE. We then evaluate existing multimodal extensions along three core design axes—position design, frequency allocation and compatibility with text-only RoPE. Through this analytical lens, we identify critical limitations in current approaches, directly motivating our proposal of two simple yet effective methods: Multi-Head RoPE and MRoPE-Interleave.

## 2.1 PRELIMINARIES: VANILLA RoPE

Vanilla RoPE (Su et al., 2024) is a pivotal method for encoding positional information in modern LLMs. Unlike additive position embeddings, RoPE applies a rotational transformation to the query and key vectors, thereby incorporating relative position dependencies directly into the self-attention mechanism. Given a query vector $q$ at position $m$ and a key vector $k$ at position $n$, the attention scores $S$ are calculated as:

$$S = (\mathcal{R}_m q)^\top (\mathcal{R}_n k) = q^\top \mathcal{R}_m^\top \mathcal{R}_n k = q^\top \mathcal{R}_{n-m} k \tag{1}$$

The transformation $\mathcal{R}$ is an orthogonal rotation, which causes the score $S$ to depend solely on the relative position $n - m$. This property is achieved by constructing $\mathcal{R}_m$ as a block-diagonal matrix parameterized by the absolute position $m$ and a set of fixed frequencies $\theta_i$. The rotation frequencies, $\theta_i = \text{base}^{-2i/d}$ for $i \in [0, d/2 - 1]$, are set according to a geometric sequence. This design creates a spectrum of frequencies ranging from high (for small $i$) to low (for large $i$), corresponding to each pair of dimensions.

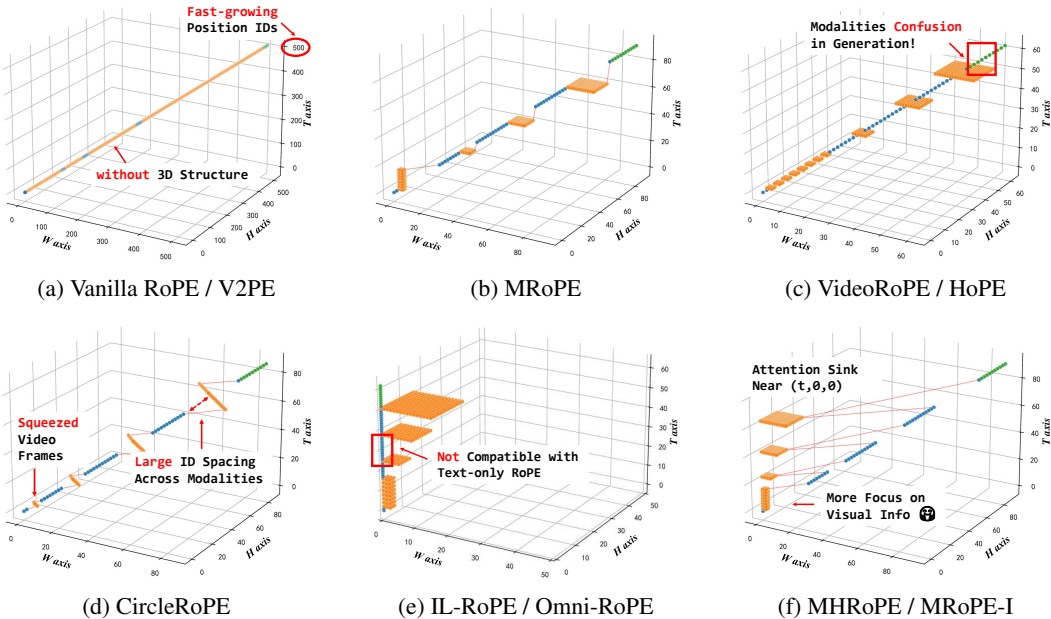

Figure 1: Position design of different multimodal RoPE variants. The illustrated example follows an interleaved multimodal sequence: `<system prompt>`, `<video 1>`, `<text>`, `<image 1>`, `<text>`, `<image 2>`, `<text>`, `<image 3>`, `<text>`, `<generated text>`.

## 2.2 POSITION DESIGN

This section governs how the positional identifier $m$ is assigned to text/visual tokens.

### 2.2.1 1D SEQUENTIAL DESIGN.

The most straightforward approach, employed by vanilla RoPE (Su et al., 2024) and V2PE (Ge et al., 2024), is to treat the multimodal input as a flattened, one-dimensional sequence. Position indices are assigned incrementally, with the position $m_i$ of the $i$-th token defined as $m_i = m_{i-1} + s_{\text{mod}}$, where $s_{\text{mod}}$ is a step size specific to the token's modality. For vanilla RoPE, all modalities are treated uniformly, with $s = 1$.

As shown in Figure 1a, this design presents two significant drawbacks. First, it discards the inherent 3D structure of visual content, which can alter the spatio-temporal reasoning capabilities of a VLM. Second, position indices can grow exceedingly large in long sequences, negatively affecting the model's extrapolation performance (Wei et al., 2025).

To address the issue of large position indices, V2PE (Ge et al., 2024) introduces dynamic position scaling for visual tokens, setting their step size $s_{\text{visual}}$ to a value in $\{1, 1/2, \ldots, 1/256\}$. This modification mitigates the rapid growth of position indices and has shown benefits in long video understanding. However, the 3D structure of visual content is still ignored in 1D sequential design.

### 2.2.2 MULTI-DIMENSIONAL DESIGN.

To preserve the native 3D structure of visual content, methods like MRoPE (Wang et al., 2024a) (See Figure 1b) extend the scalar position identifier to a multi-dimensional tuple. For instance, a token's position can be represented as $\boldsymbol{m}_i = (m_i^t, m_i^h, m_i^w)$, corresponding to its temporal, vertical, and horizontal axes. MRoPE conceptually treats each visual content (e.g., an image or a set of video frames) as a single, large "cube". The temporal position of the subsequent token is then set by "jumping" past the maximum coordinate value of current block. This is achieved with the update rule:

$$m_{\text{next}}^t = \max(m_{\text{prev}}^t, m_{\text{prev}}^h, m_{\text{prev}}^w) + 1 \tag{2}$$

This strategy guarantees that no positional overlaps occur between modalities. However, proponents of VideoRoPE (Li et al., 2025) and HoPE (Li et al., 2025) argue that MRoPE's position design lacks inter-modal symmetry, they introduce a "diagonal layout" by centering the spatial coordinates (see Figure 1c). In this scheme, visual frames are not only stacked along the temporal axis but are also shifted along the vertical and horizontal axes. Despite its theoretical elegance, this diagonal layout introduces a critical flaw: the potential for position id overlap between visual content and generated text tokens. For high-resolution image content like documents, the spatial coordinates of visual tokens can extend into the index range subsequently assigned to the generated text tokens. We identify this positional ambiguity as a source of "modalities confusion in generation", a failure mode that manifested as endless text repetition in our later experiments.

CircleRoPE (Wang et al., 2025) arranges image tokens in a circular layout, orthogonal to the linear axis of text positions (see Figure 1d). A key property of this design is that it renders all visual tokens equidistant from any given text token, which theoretically promotes uniform attention across the image. However, CircleRoPE's design has two limitations. First, the large interval between modalities may impede effective cross-modal interaction. Second, lacking a temporal axis, it collapses all video frames onto a single ring, which introduces severe temporal ambiguity.

### 2.2.3 TOWARDS AN OPTIMAL POSITION DESIGN

Our preceding analysis, summarized in the first column of Table 1, indicates that a robust position design must satisfy several criteria, which we collectively term Positional Coherence: (i) preserve the 3D structure of visual content; (ii) maintain a slow growth rate; (iii) avoid modalities confusion in generation; (iv) establish an appropriate modality interval.

While MRoPE fulfills most of these requirements, our analysis uncovers a crucial phenomenon: MRoPE exhibits a visual "attention sink", where attention concentrates on the top-left corner of each image or video frame, a behavior visualized in Figure 2. Specifically, for the image input from ChartQA, it is duplicated and paired with the prompt "Describe the two images in detail" to reveal the attention sink in an interleaved pattern. This phenomenon is analogous to the attention sink observed at the initial tokens in large language models. This insight directly motivates our proposal of *spatial-reset*: a mechanism that resets the spatial dimensions for each visual content. By applying *spatial-reset*, we aim to align this visual sink with the LLM's bias for small position IDs, accelerating visual adaptation.

Furthermore, *spatial-reset* provides another benefit for video understanding by disentangling the representation of motion. Consider an object token at spatial coordinates $(h_1, w_1)$ at time $t_1$ and a second token for the same object at $(h_2, w_2)$ at time $t_2$. Let their absolute position indices be $\boldsymbol{m}_1$ and $\boldsymbol{m}_2$, respectively. Under the standard MRoPE formulation, the temporal and spatial dimensions are coupled. The absolute positions are $\boldsymbol{m}_1 = (t_1, t_1 + h_1, t_1 +$

Figure 2: Visual attention sink in MRoPE. Average attention scores to input sequence from the ChartQA (Left) and VideoMME-short (Right).

$w_1$) and $\boldsymbol{m}_2 = (t_2, t_2 + h_2, t_2 + w_2)$. The resulting relative position indices, $\boldsymbol{m}_{\mathrm{rel}} = \boldsymbol{m}_2 - \boldsymbol{m}_1$, becomes entangled:

$$\boldsymbol{m}_{\mathrm{rel}} = (t_2 - t_1, (t_2 - t_1) + (h_2 - h_1), (t_2 - t_1) + (w_2 - w_1)) \tag{3}$$

In contrast, our method with *spatial-reset* decouples these dimensions. The positions are defined as $\boldsymbol{m}_1 = (t_1, h_1, w_1)$ and $\boldsymbol{m}_2 = (t_2, h_2, w_2)$. This yields a purely spatio-temporal relative vector:

$$\boldsymbol{m}'_{\mathrm{rel}} = (t_2 - t_1, h_2 - h_1, w_2 - w_1) \tag{4}$$

This disentangled representation of motion is more intuitive and provides a cleaner inductive bias for the model to learn from. Therefore, the position design we adopt for our proposed MHRoPE and MRoPE-I methods builds upon MRoPE by incorporating *spatial-reset* (as illustrated in Figure 1f).

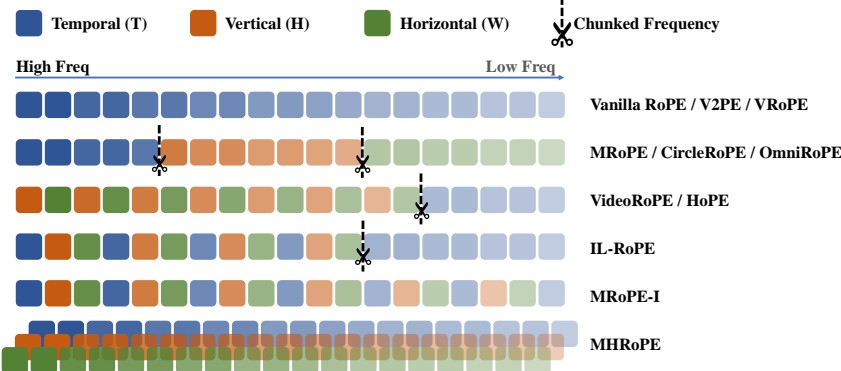

Figure 3: Frequence allocation of different multimodal RoPEs.

## 2.3 FREQUENCY ALLOCATION

Frequency Allocation governs the assignment of feature channels and their corresponding frequencies $\theta_i$ to the various axes of the position identifier $\boldsymbol{m}$ (temporal $t$, vertical $h$ or horizontal $w$).

### 2.3.1 FREQUENCY ALLOCATION IN 1D RoPE.

In 1D methods like vanilla RoPE and V2PE, all feature channels are allocated to encode the temporal axis. The frequencies $\theta_i$ decay as the channel index $i$ increases, creating a spectrum from high-frequency (for short-range dependencies) to low-frequency (for long-range dependencies). This design also imparts a long-range decay property on attention scores, as their upper bound is a function of the relative distance, which can be approximated by $\sum_{i=0}^{d/2-1} |S_{i+1}|$, where $S_j = \sum_{k=0}^{j-1} e^{\mathrm{i}(m-n)\theta_k}$ (see Appendix D.2 for detailed derivation). V2PE's position scaling for visual tokens effectively slows this decay, enhancing the model's ability to focus on long visual content.

### 2.3.2 MULTI-DIMENSIONAL FREQUENCY ALLOCATION.

The standard MRoPE partitions the $d$ feature dimensions into three contiguous blocks, dedicating one to each of the $t, h,$ and $w$ axes. As rotational frequencies decrease with channel index, this design forces the temporal axis to be encoded entirely by the highest-frequency channels. This creates a strong inductive bias that is detrimental to long-sequence understanding, as it leads to a rapid decay of attention over time. Furthermore, because the $h$ and $w$ axes are assigned distinct, non-overlapping frequency ranges, they exhibit different long-range decay rates, as visualized in Figure 4a. This asymmetry can impair the model's ability to learn consistent spatial relationships.

Subsequent methods have attempted to rectify this temporal bias through various frequency re-allocation strategies. VideoRoPE and HoPE, for example, move the temporal axis to occupy the low-frequency channels. IL-RoPE employs a form of interleaving but similarly reserves the lowest-frequency channels for the temporal dimension. While these approaches can mitigate the long-context issue for the temporal axis, they introduce a critical, unaddressed trade-off: they force the spatial dimensions into a restricted, and often exclusively high-frequency, band. This severely limits

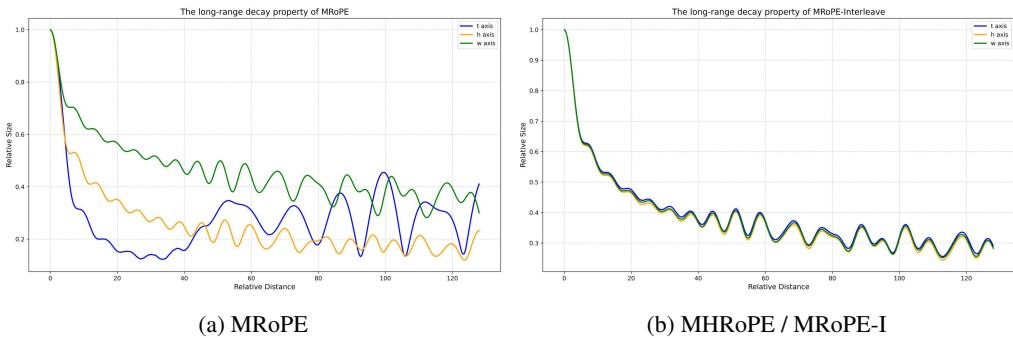

Figure 4: The long-range decay property of MRoPE, MHRoPE and MRoPE-I.

the model's ability to capture multi-scale spatial relationships, which can impair performance on tasks reliant on fine-grained spatial reasoning, such as visual grounding. Furthermore, the very act of partitioning feature dimensions inherently coarsens the frequency resolution for each positional axis. The performance implications of this reduced granularity is under-exploration.

### 2.3.3 TOWARDS AN OPTIMAL FREQUENCY ALLOCATION

To address the limitations of frequency allocation, we propose two effective strategies, as summarized in the second column of Table 1. Both methods resolve the rapid temporal decay and asymmetric spatial decay of MRoPE, yielding a unified decay profile for all axes as shown in Figure 4b.

**Multi-Head Allocation.** The first strategy, termed Multi-Head RoPE (MHRoPE), is inspired by recent work demonstrating channel-level redundancy in RoPE (e.g., partial RoPE (Barbero et al., 2025)). Based on the premise that similar redundancy exists at the attention head level, MHRoPE partitions the positional encoding task among different attention heads[1], as shown in Figure 3. A primary advantage of this strategy is that each axis is encoded using the full frequency spectrum available within its assigned heads. This approach avoids the loss of frequency resolution inherent to channel-splitting methods. Moreover, it may be more scalable. As the number of positional axes grows (Liu et al., 2025), partitioning a fixed channel channels (e.g., 128) becomes untenable, whereas dedicating distinct heads to new dimensions offers a far more robust and flexible approach.

**Interleaved Allocation.** Our second strategy, employed in our MRoPE-Interleaved (MRoPE-I) method, distributes feature channels to the temporal ($t$), vertical ($h$), and horizontal ($w$) axes in a fine-grained, round-robin manner, as shown in Figure 3. This design ensures that each positional axis is encoded using the full frequency spectrum, from high to low, thereby enabling robust multi-scale modeling for each positional axis. Moreover, the uniform frequency distribution of our interleaved design is compatible with extrapolation algorithms like NTK-aware (bloc97, 2023) and YaRN (Peng et al., 2024), which function by rescaling the frequency spectrum (see Appendix D.3).

### 2.4 COMPATIBILITY WITH TEXT-ONLY ROPE.

Most VLMs are adapted from LLMs, which typically use vanilla RoPE for positional encoding. This raises a natural question: should the encoding strategy for text tokens in VLMs remain identical to that of the base LLM? While most works implicitly agree, some methods have explored deviations.

From the Position Design aspect, methods like IL-RoPE (Liao et al., 2025) and Omni-RoPE (Wu et al., 2025) modify the text encoding. As shown in Figure 1e, they reset spatial coordinates for each image to aid editing but concurrently set the spatial dimensions for text tokens to zero. This design choice breaks compatibility with the standard RoPE used in the pre-trained LLM.

From the Frequency Allocation aspect, we also explored a potential modification. Since the coordinate range of spatial dimensions is much smaller than that of the temporal axis, a smaller rotary 'base' could be used to better encode the spatial dimension. However, our experiments showed

---

[1]For Group Query Attention (GQA), we partition on KV heads and repeat on corresponding query heads.

that this strategy led to poor performance across most benchmarks. This outcome strongly indicates the critical importance of maintaining full compatibility with the text-only RoPE for effective knowledge transfer from pre-trained LLMs.

## 2.5 PROPOSED MULTIMODAL ROPES

Based on our analysis, we propose two novel multimodal RoPE variants: Multi-Head RoPE (MHRoPE) and MRoPE-Interleave (MRoPE-I). Both methods are built upon a shared set of design guidelines for robustness and performance.

For position design, we enhance MRoPE position design with *spatial-reset* to improve the model's focus on visual information. We also maintain strict compatibility with text-only RoPE, ensuring the effective transfer of pre-trained knowledge.

The key distinction between our variants lies in their frequency allocation strategy. MHRoPE employs Multi-Head Allocation, dedicating distinct attention heads to different axes to preserve frequency resolution and offer scalability. In contrast, MRoPE-I uses Interleaved Allocation, a fine-grained approach ensuring full-spectrum encoding and compatibility with extrapolation techniques. For a detailed discussion on the trade-offs between MHRoPE and MRoPE-I, see Appendix D.1.

## 3 EXPERIMENT

### 3.1 EXPERIMENTAL SETUP

**Training Details.** All models use the QwenViT and connector from Qwen2.5VL[2], while initializing the VLM backbone with the Qwen2.5 7B LLM. During training, we freeze the ViT to fix the visual representations and unfreeze the connector and LLM backbone. This strategy is designed to isolate the effects of our proposed RoPE modifications while adhering to the standard VLM adaptation paradigm of building upon a pre-trained LLM. Notably, this initialization setting aligns with that of VideoRoPE (Wei et al., 2025) and HoPE (Li et al., 2025), and differs from alternative architectural explorations in VLMs such as Apollo (Orr Zohar & Xia, 2024) and CamBrain-1 (Tong et al., 2024).

The training process adopts a batch size of 128 and uses the AdamW optimizer with $\alpha$=0.9, $\beta$=0.98, and weight decay of 0.05. The learning rate follows a cosine decay schedule from an initial value of $1 \times 10^{-5}$ down to a minimum of $3 \times 10^{-6}$. Each experiment consumes approximately 512 NVIDIA A100 GPU hours. The training context length is set to 32K, and the rotary base is set to 1000000. All experiments share identical training data, model architecture, and hyperparameters, with the sole difference being the choice of multimodal RoPE.

**Training Data and Evaluation Benchmarks.** We conduct experiments using approximately 2M high-quality supervised fine-tuning (SFT) samples, covering a wide range of visual scenarios including image captioning, OCR, visual reasoning, visual grounding, document comprehension, long video understanding and multi-turn dialogue. For evaluation, we adopt more than 20 benchmarks spanning images, videos, and grounding tasks. Specifically, the image benchmarks include MMMU (Yue et al., 2024), MMBench (Liu et al., 2024a), MMStar (Chen et al., 2024), OCRBench (Liu et al., 2024b), AI2D (Kembhavi et al., 2016), RealWorldQA (X.AI, 2024), DocVQA (Mathew et al., 2021), TextVQA Singh et al. (2019), InfoVQA (Mathew et al., 2022), and ChartQA (Masry et al., 2022). To evaluate multi-image reasoning capabilities, we use the BLINK benchmark (Fu et al., 2024). The video benchmarks consist of MVBench (Li et al., 2024), STAR (Wu et al., 2021), VideoMME (Fu et al., 2025), LVBench (Wang et al., 2024b), MLVU (Zhou et al., 2024b), and Charades-STA (Zhou et al., 2024a). For grounding, we evaluate on RefCOCO (Kazemzadeh et al., 2014) series.

**Evaluation Setup.** All results are reported following the official evaluation protocol of Qwen2VL (Bai et al., 2025). Since all models are trained with a 32K context length, we cap the maximum number of visual tokens at 24K and apply "smart resize" to preserve dynamic visual resolution. For video inputs, we sample frames at 2 FPS by default; when this yields more than 768 frames, we fall back to uniform sampling to retain at most 768 frames.

---

[2]For fair comparison with prior work which using Qwen2VL, we disable the absolute time encoding used in Qwen2.5VL.

Table 2: Overall performance of multimodal RoPEs variants on various benchmarks. The highest score is shown in **bold**, while the second-highest score is underlined.

| Types | Benchmarks | Vanilla RoPE | MRoPE | VideoRoPE | HoPE | CircleRoPE | MHRoPE | MRoPE-I |
|---|---|---|---|---|---|---|---|---|
| Image | MMMU | 50.56 | 50.22 | 49.89 | 49.89 | 47.22 | 53.00 | **53.22** |
| | MMBench$_{avg}$ | 74.75 | 74.06 | **75.95** | 75.35 | 74.91 | 75.04 | 75.56 |
| | MMstar | 49.13 | 49.93 | 49.60 | 50.33 | 47.00 | 49.60 | **51.13** |
| | OCRBench | 73.40 | 72.70 | 66.20 | 66.60 | 70.60 | 73.40 | **74.00** |
| | AI2D | **76.20** | 74.94 | 74.29 | 76.10 | 74.45 | 75.45 | 75.36 |
| | RealworldQA | 58.30 | 57.25 | 56.21 | 57.12 | 56.60 | **60.52** | 57.39 |
| | DocVQA | 82.94 | 81.49 | 60.13 | 60.12 | 77.70 | 81.32 | **83.72** |
| | TextVQA | 66.80 | 65.85 | 66.58 | 66.77 | 65.54 | 66.49 | **66.91** |
| | InfoVQA | **58.85** | 52.96 | 37.42 | 34.80 | 53.11 | 52.01 | 58.24 |
| | ChartQA | 56.84 | **63.56** | 54.88 | 55.44 | 53.72 | 62.44 | 62.12 |
| | BLINK | 36.12 | 37.93 | 36.20 | 35.98 | 40.88 | 42.80 | **44.08** |
| Video | MVBench | 57.05 | 57.85 | 56.78 | 58.00 | 57.10 | **58.93** | 57.05 |
| | STAR | 58.07 | 58.28 | 57.20 | 58.30 | 57.94 | **59.48** | 57.79 |
| | MLVU | 64.69 | 63.26 | **66.05** | 64.81 | 62.69 | 65.69 | 65.46 |
| | VideoMME | 58.63 | 58.22 | 58.70 | **59.52** | 57.70 | 57.48 | 58.96 |
| | LVBench | 38.93 | 39.22 | 40.15 | **40.99** | 38.80 | 40.32 | 40.54 |
| | Charades-STA | 32.49 | 32.23 | 34.21 | **36.07** | 32.27 | 33.56 | 34.36 |
| Grounding | RefCOCO$_{val}$ | 77.67 | 78.35 | 77.95 | 77.72 | 79.59 | 79.87 | **80.94** |
| | RefCOCO$_{testA}$ | 81.37 | 82.52 | 80.43 | 81.60 | 83.98 | 83.66 | **84.55** |
| | RefCOCO$_{testB}$ | 72.66 | 72.31 | 72.62 | 71.44 | 74.35 | 73.20 | **75.05** |
| | RefCOCO+$_{val}$ | 69.16 | 68.80 | 68.15 | 69.61 | 70.19 | 70.55 | **71.80** |
| | RefCOCO+$_{testA}$ | 74.48 | 75.95 | 73.14 | 74.55 | 76.77 | 76.88 | **77.44** |
| | RefCOCO+$_{testB}$ | 61.67 | 59.97 | 60.50 | 61.69 | **62.59** | 61.96 | 61.96 |
| | RefCOCOg$_{val}$ | 75.45 | 75.86 | 74.06 | 74.69 | 76.10 | 76.55 | **77.70** |
| | RefCOCOg$_{test}$ | 75.40 | 75.73 | 73.90 | 75.45 | 76.12 | 76.68 | **77.34** |
| Overall | Image | 62.17 | 61.90 | 57.03 | 57.14 | 60.16 | 62.92 | **63.79** |
| | Video | 51.64 | 51.51 | 52.18 | **52.95** | 51.09 | 52.58 | 52.36 |
| | Grounding | 73.48 | 73.69 | 72.59 | 73.34 | 74.96 | 74.92 | **75.85** |

## 3.2 OVERALL PERFORMANCE

The overall performance of different multimodal RoPEs is presented in Table 2. Both MHRoPE and MRoPE-I achieve consistently better performance across the majority of benchmarks. For instance, MRoPE-I outperforms the vanilla RoPE baseline by a significant margin of $+2.67\%$ on MMMU, $+5.28\%$ on ChartQA, and $+3.27\%$ on RefCOCO$_{val}$.

The results also reveal that while vanilla RoPE serves as a competitive baseline, its performance is noticeably impaired on benchmarks that demand fine-grained spatial reasoning, such as ChartQA and the RefCOCO series. This performance gap highlights the fundamental limitations of its flattened, 1D position design. Vanilla RoPE also suffers from extrapolation, see Appendix D.4.

While VideoRoPE and HoPE demonstrate stronger performance on video benchmarks, they exhibit anomalous degradation on DocVQA, InfoVQA, and ChartQA. We attribute this discrepancy to a critical flaw in their position design: the overlap of position indices, which induces confusion between visual and generated text tokens. The ablation study in Table 4 confirms that this confusion is the root cause of the degradation.

The suboptimal designs of MRoPE and CircleRoPE manifest in their performance. MRoPE breaks the full frequency spectrum for each positon axis. Consequently, it struggles on tasks demanding specific frequency ranges, such as long-video understanding (MLVU, LVBench), which requires robust low-frequency temporal encoding, and visual grounding (RefCOCO), which benefits from high-frequency spatial encoding. Similarly, CircleRoPE introduces a large modality interval and collapses the video positions, results in poor video understanding.

In contrast, MHRoPE and MRoPE-I leverage *spatial-reset* position design, which prevents modalities confusion and not introducing an improper modality interval. By providing each positional axis with a full frequency spectrum (interleave or multi-head allocation), they enable the models to better capture both fine-grained spatial details (high-frequency) and long-range temporal dependencies (low-frequency), leading to their superior overall performance.

### 3.3 GENERALIZATION ACROSS ARCHITECTURES

Current Vision-Language Models (VLMs) are converging towards a unified architectural paradigm comprising a vision encoder, a connector, and a Large Language Model (LLM). Since multimodal positional encodings primarily operate within the LLM backbone, which exhibits minimal structural variation across different VLM families. We hypothesize that our proposed MHRoPE and MRoPE-I possess strong generalizability.

To empirically validate this hypothesis, we extended our evaluation to Qwen3-VL-4B-Instruct (QwenVL Team, 2025) and Qwen3-VL-8B-Instruct. Although they belong to the same lineage as Qwen2.5-VL, they feature significant architectural distinctions: (1) the removal of window attention in the vision encoder; (2) the introduction of a DeepStack architecture between the connector and the backbone; (3) the application of QK-Norm within the LLM backbone; and (4) specific to the 4B variant, the tying of weights between the embedding layer and the LM head.

The results are presented in Table 3 (with full details in Appendix D.6). Despite these architectural shifts, our methods consistently achieve the best performance. Furthermore, previously observed phenomena, such as the performance degradation caused by diagonal layouts, are corroborated in these new experiments, further solidifying the validity and robustness of our approach.

Table 3: Average performance across modalities for RoPE variants on Qwen3-VL-4B and Qwen3-VL-8B Instruct models. Highest scores are in **bold**, second-highest are underlined. Grnd: average grounding score on RefCOCO benchmarks.

(a) Qwen3-VL-4B-Instruct

| Method | Image | Video | Grnd |
|---|---|---|---|
| RoPE | 47.00 | 49.90 | 20.77 |
| MRoPE | 46.67 | 49.60 | 23.67 |
| VideoRoPE | 42.77 | 49.86 | 18.91 |
| HoPE | 43.20 | 50.25 | 18.74 |
| CircleRoPE | 47.36 | 49.23 | 18.84 |
| MHRoPE | 48.22 | **50.80** | 25.23 |
| MRoPE-I | **48.82** | 50.49 | **27.52** |

(b) Qwen3-VL-8B-Instruct

| Method | Image | Video | Grnd |
|---|---|---|---|
| RoPE | 63.35 | 56.78 | 70.09 |
| MRoPE | 62.95 | 57.03 | 70.97 |
| VideoRoPE | 60.26 | 57.75 | 67.99 |
| HoPE | 59.96 | **57.81** | 70.19 |
| CircleRoPE | 63.54 | 54.88 | 71.20 |
| MHRoPE | 64.63 | 57.46 | 73.03 |
| MRoPE-I | **64.82** | 57.64 | **75.46** |

### 3.4 ABLATION STUDY

This section presents ablation studies on key design choices for our robust multimodal RoPEs. Additional results are provided in Appendix D.5.

#### 3.4.1 ABLATION STUDY ON POSITION DESIGN

We previously argued that an optimal position design should: (1) incorporate 3D structure to capture native spatio-temporal information, (2) maintain a proper modality interval, and (3) preserve compatibility with text-only RoPE. To systematically dissect the impact of these factors, we conduct an ablation study, fixing the frequency allocation strategy to our interleaved allocation while varying the position design. The results are presented in Table 4. Simply introducing a 3D structure over the vanilla RoPE provides a notable boost to grounding performance. The addition of *spatial-reset* mechanism yields substantial gains across all benchmark categories, confirming its effectiveness. We also ablated other position designs proposed in prior work, as shown in Table 4.

**Diagonal Layout:** Implementing the diagonal layout from VideoRoPE leads to a severe degradation in performance on document-centric benchmarks (DocVQA, InfoVQA and ChartQA). A qualitative analysis reveals a specific failure mode: repetitive, nonsensical text generation (e.g., "1111..."), which occurs even when the layout is applied only at inference time. We attribute this behavior to modalities confusion induced by positional overlap, causing the model to misinterpret its own generated text tokens as visual tokens, resulting in this unpredictable repetitive output.

**Enlarged Modality Interval:** We also tested artificially enlarging the modality interval to match that of vanilla RoPE, a strategy similar to RoPE-Tie (Su, 2024). This also resulted in poor document-related performance. However, the failure mode was distinct: the model generated fluent but contex-

tually irrelevant text, effectively ignoring the visual input. This suggests that while a clear modality interval is necessary, simply maximizing its size to align with vanilla RoPE can be detrimental by

**Text *spatial-reset*.** We also tested the strategy from IL-RoPE and Omni-RoPE, which resets spatial dimensions for visual tokens as well as text ones (Fig. 1e). This approach resulted in a notable performance degradation compared to the vanilla RoPE, emphasizing that preserving RoPE alignment for text is critical for successfully adapting LLMs into VLMs.

**Scaling rotary base.** Motivated by the smaller coordinate range of the spatial axes, we experimented with scaling their corresponding rotary base (e.g., from 1,000,000 to 10,000). This consistently resulted in a clear performance drop on image benchmarks. This finding demonstrates that even well-intentioned deviations from the base LLM's RoPE formulation can break compatibility and severely impair knowledge transfer.

| Position Design | Image | Grounding | Video | DocVQA | InfoVQA | ChartQA |
|---|---|---|---|---|---|---|
| vanilla RoPE | 65.69 | 73.48 | 51.64 | 82.94 | **58.85** | 56.84 |
| + 3D structure | 65.87 | 74.40 | 51.29 | 82.33 | 57.24 | 61.44 |
| + 3D + *spatial-reset* | **66.65** | **75.85** | 52.36 | **83.72** | 58.24 | **62.12** |
| + diagonal layout | 61.20 | 72.33 | **52.51** | 60.13 | 37.42 | 54.88 |
| + modality interval | 62.80 | 73.19 | 50.88 | 70.43 | 42.18 | 51.28 |
| + text *spatial-reset* | 58.27 | 68.2 | 50.71 | 77.30 | 52.15 | 44.33 |
| + scaling rotary base | 60.15 | 74.13 | 52.11 | 80.44 | 52.16 | 58.80 |

Table 4: Ablation study of different position design strategies.

### 3.4.2 Ablation Study on Frequency Allocation

To determine the optimal frequency allocation strategy, we fix the position design the same as MRoPE with *spatial-reset* enhancement, and vary only the frequency allocation scheme. As shown in Table 5, a more uniform allocation strategy consistently outperforms alternatives that split the spectrum into partial chunks. This highlights the importance of ensuring that each positional axes (time, height, width) retains access to the full frequency spectrum.

| Allocation Type | Image | Video | Grounding | Overall |
|---|---|---|---|---|
| VideoRoPE-like | 65.33 | 52.11 | 72.50 | 63.31 |
| IL-RoPE-like | 65.26 | 51.15 | 72.80 | 63.07 |
| Multi-Head | 66.40 | 52.58 | 74.92 | 64.63 |
| Interleave | 66.65 | 52.36 | 75.85 | **64.95** |

Table 5: Ablation results of different frequency allocation strategies.

## 4 Conclusion

In this work, we conducted the first systematic investigation into multimodal Rotary Positional Embedding (RoPE) for Vision-Language Models (VLMs). From our systematic comparison and extensive experiments, we identified three key design considerations for robust multimodal RoPE: positional coherence, full frequency utilization, and preservation textual priors from pre-trained LLMs. Guided by these insights, we proposed two plug-and-play RoPE variants: Multi-Head RoPE (MHRoPE) and MRoPE-Interleave (MRoPE-I). Both methods adhere to our identified guidelines, effectively addressing common failure modes and achieving significant performance in both general and fine-grained multimodal understanding. This work offers a comprehensive guide for designing effective multimodal positional encodings, paving the way for future advancements in VLMs.

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

## A  ETHICS STATEMENT

This work adheres to the ICLR Code of Ethics. In this study, no human subjects or animal experimentation was involved. All datasets used were sourced in compliance with relevant usage guidelines, ensuring no violation of privacy. We have taken care to avoid any biases or discriminatory outcomes in our research process. No personally identifiable information was used, and no experiments were conducted that could raise privacy or security concerns. We are committed to maintaining transparency and integrity throughout the research process.

## B    Reproducibility Statement

We have made every effort to ensure that the results presented in this paper are reproducible. Code will be made publicly available to facilitate replication and verification after inspection. The experimental setup, including training steps, model configurations, and hardware details, is described in detail in the paper. We believe these measures will enable other researchers to reproduce our work and further advance the field.

## C    LLM Usage

Large Language Models (LLMs) were used to aid in the writing and polishing of the manuscript. Specifically, we used an LLM to assist in refining the language, improving readability, and ensuring clarity in various sections of the paper. The model helped with tasks such as sentence rephrasing, grammar checking.

It is important to note that the LLM was not involved in the ideation, research methodology, or experimental design. All research concepts, ideas, and analyses were developed and conducted by the authors. The contributions of the LLM were solely focused on improving the linguistic quality of the paper, with no involvement in the scientific content or data analysis.

The authors take full responsibility for the content of the manuscript, including any text generated or polished by the LLM. We have ensured that the LLM-generated text adheres to ethical guidelines and does not contribute to plagiarism or scientific misconduct.

## D    Appendix

### D.1    Practical Considerations: MHRoPE vs. MRoPE-I

While both of our proposed methods are effective, we currently recommend MRoPE-I over MHRoPE for two primary reasons: its consistent (albeit slight) performance advantage and its greater implementation simplicity. We attribute MHRoPE's minor performance deficit to its head-level information partitioning, which prevents the integration of different positional axes within the self-attention mechanism. From an engineering perspective, MRoPE-I is also simpler, avoiding the complexities that MHRoPE introduces with distributed training paradigms like tensor parallelism. Nevertheless, MHRoPE's design offers a potentially more scalable architecture for future models that may need to accommodate a larger number of positional axes.

### D.2    Derivation of the Attention Score Upper Bound in MRoPE

Here, we provide a formal derivation for the upper bound of the RoPE attention score. The RoPE dot product between a query $q$ and a key $k$ at a relative position $m - n$ can be expressed in complex form as:

$$(\mathcal{R}_m q)^\top (\mathcal{R}_n k) = \mathrm{Re}\left[ \sum_{i=0}^{d/2-1} (q_{[2i:2i+1]} \cdot k^*_{[2i:2i+1]}) e^{\mathrm{i}(m-n)\theta_i} \right] \tag{5}$$

where $v^*$ denotes the complex conjugate of a 2D vector treated as a complex number, and $\cdot$ is the complex product.

To derive the upper bound, we analyze the magnitude of the summation term. To apply summation by parts, let us define a content-dependent sequence $h_i = q_{[2i:2i+1]} \cdot k^*_{[2i:2i+1]}$ and a position-dependent sequence of partial sums $S_j = \sum_{k=0}^{j-1} e^{\mathrm{i}(m-n)\theta_k}$. We also set the boundary conditions $S_0 = 0$ and $h_{d/2} = 0$. The standard summation by parts formula is $\sum_{i=a}^{b} u_i \Delta v_i = [u_i v_i]_a^{b+1} - \sum_{i=a}^{b} v_{i+1} \Delta u_i$. Applying this, the magnitude of the summation can be rewritten and bounded as

follows:

$$
\begin{aligned}
\left| \sum_{i=0}^{d/2-1} h_i e^{\mathrm{i}(m-n)\theta_i} \right| &= \left| [h_i S_i]_0^{d/2} - \sum_{i=0}^{d/2-1} S_{i+1}(h_{i+1} - h_i) \right| \\
&= \left| (h_{d/2} S_{d/2} - h_0 S_0) - \sum_{i=0}^{d/2-1} S_{i+1}(h_{i+1} - h_i) \right| \\
&= \left| - \sum_{i=0}^{d/2-1} S_{i+1}(h_{i+1} - h_i) \right| \\
&\leq \sum_{i=0}^{d/2-1} |S_{i+1}||h_{i+1} - h_i| \\
&\leq \left( \max_{0 \leq i < d/2} |h_{i+1} - h_i| \right) \sum_{i=0}^{d/2-1} |S_{i+1}|
\end{aligned}
\tag{6}
$$

This final expression reveals that the upper bound is a product of two distinct components. $\max |h_{i+1} - h_i|$ is a content-dependent term that acts as a scaling factor based on the specific query and key vectors. The second, $\sum |S_{i+1}|$, is a purely position-dependent term whose value is determined only by the relative position $m - n$ and the fixed frequencies $\theta_i$. Since the content-dependent term is independent of position, the long-range decay property of the attention score is governed primarily by this position-dependent term. Therefore, its average value, $\frac{1}{d/2} \sum_{i=1}^{d/2} |S_i|$, serves as a practical indicator to characterize how the upper bound attenuates with relative distance.

### D.3 COMPATIBILITY WITH YARN EXTRAPOLATION

As shown in Figure 4b, the interleaved frequency allocation of MRoPE-I makes it compatible with extrapolation algorithms like NTK-aware (bloc97, 2023) and YaRN (Peng et al., 2024). Whereas standard MRoPE's partitioned spectrum complicates the application of a consistent frequency scaling boundary, our interleaved design provides a full spectrum across all positional axes, enabling a straightforward and symmetric application of these methods.

To validate this effect, we apply YaRN to both MRoPE and MRoPE-I under a 256K context window and evaluate their performance on LVBench and MLVU. As shown in Table 6, MRoPE-I with YaRN demonstrates substantially larger gains in long-video understanding compared to MRoPE.

Table 6: Performance comparison of MRoPE and MRoPE-I with and without YaRN under a 256K context.

| Method | LVBench | MLVU |
|---|---|---|
| MRoPE | 41.5 | 62.9 |
| MRoPE + YaRN | 41.2 | 63.3 |
| MRoPE-I | 42.0 | 63.2 |
| MRoPE-I + YaRN | **43.6** | **64.1** |

### D.4 LONG-CONTEXT VIDEO UNDERSTANDING

We further compare the performance of different methods on long video understanding, with context lengths ranging from 32K to 256K. As shown in Figure 5, apart from LVBench, we do not observe clear performance improvements or degradation when extrapolating to longer sequences. The only exception is Vanilla RoPE, which suffers from a sharp performance drop at 128K/256K. We attribute this to excessively fast-growing position IDs, which lead to degraded extrapolation capability, which also discussed in other works (Wei et al., 2025; Li et al., 2025).

Overall, methods such as VideoRoPE and HoPE, which allocate most low-frequency channels to the temporal axis, exhibit slightly better extrapolation ability in long video senario. However, when

considering performance across images and grounding tasks, MHRoPE and MRoPE-I remain the most comprehensive and balanced designs.

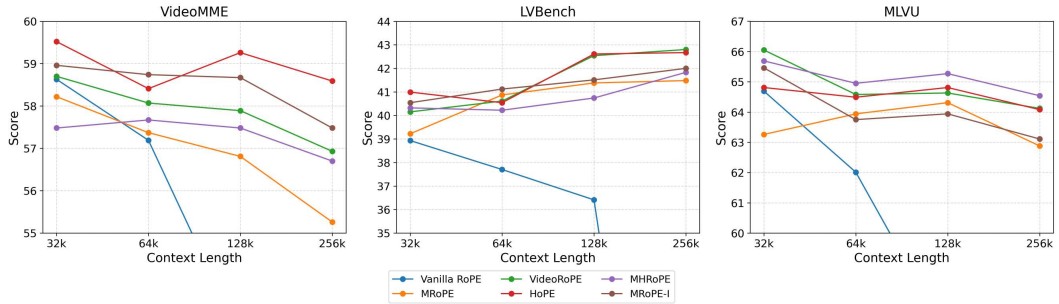

Figure 5: Video extrapolation performance. Models are trained with a context length of 32k (256 frames) and extrapolated to 64k (512 frames), 128k (1024 frames), and 256k (2048 frames).

## D.5 MORE ABLATION RESULTS.

### D.5.1 ENHANCED VISUAL ATTENTION IN *spatial-reset*

To understand the mechanism driving the effectiveness of *spatial-reset*, we analyzed its impact on the model's attention patterns. As detailed in Table 7, we calculated the total attention scores on visual tokens using the DocVQA test set. Specifically, we extracted attention scores from layers 4, 12, 20, and 28, and averaged the scores across all attention heads and samples. The result demonstrates that MRoPE equipped with *spatial-reset* allocate more attention on visual content, particularly in deeper layers, confirming it's effectiveness in enhancing the model's visual focus.

| Method | Layer 4 | Layer 12 | Layer 20 | Layer 28 |
|---|---|---|---|---|
| MHRoPE | 40.31 | 21.76 | 32.05 | 19.00 |
| w/o *spatial-reset* | 35.99 | 19.68 | 22.02 | 9.93 |
| MRoPE-I | 37.48 | 15.68 | 28.08 | 23.23 |
| w/o *spatial-reset* | 31.22 | 17.66 | 16.02 | 11.69 |

Table 7: Average attention scores (%) on visual contents. The inputs are from DocVQA test set. And the scores are averaged between attention heads and samples.

### D.5.2 ALLOCATION RATIO OF FREQUENCY

We further investigate different allocation ratios under the interleave frequency strategy. The results are summarized in Table 8. The balanced allocation (t:h:w = 24:20:20) achieves the best overall performance. Increasing the proportion of channels assigned to the temporal axis reduces the available high-frequency capacity for spatial dimensions. This leads to a degradation in grounding ability and negatively impacts benchmarks involving spatial understanding in both images and videos.

| Allocation Ratio | Image | Video | Grounding | Overall |
|---|---|---|---|---|
| 0:32:32 | 66.42 | 51.01 | 76.02 | 64.48 |
| 12:26:26 | 66.30 | 51.93 | 75.77 | 64.67 |
| 24:20:20 | 66.65 | 52.36 | 75.85 | **64.95** |
| 32:16:16 | 64.07 | 51.15 | 74.65 | 63.29 |
| 48: 8: 8 | 65.06 | 51.17 | 72.87 | 63.03 |

Table 8: Ablation results of different frequency allocation ratios under interleave design.

### D.5.3 TEMPORAL STRIDE IN VIDEO MODELING

This section investigates the impact of different temporal strides between video frames. Specifically, we experiment with $\delta = 0.5, 1, 2$, as well as dynamic strides as used in V2PE and HoPE (with $\delta = 1$ applied during inference). The results are shown in Table 9.

| Stride | MVBench | STAR | VideoMME | LVBench | MLVU | Charades | **Overall** |
|--------|---------|------|----------|---------|------|----------|-------------|
| 0.5 | 56.55 | 57.90 | 58.96 | 38.99 | 62.37 | 31.88 | 51.11 |
| 1 | **57.05** | 57.79 | 58.96 | 40.54 | **65.46** | **34.36** | **52.36** |
| 2 | 55.70 | **58.13** | 58.15 | 38.02 | 63.11 | 33.51 | 51.10 |
| Dynamic | 56.28 | 57.93 | 58.74 | **41.12** | 63.75 | 32.99 | 51.80 |

Table 9: Comparison of temporal stride settings for video benchmarks on MRoPE-I.

From the results, $\delta = 1$ achieves the best overall performance, while smaller ($\delta = 0.5$) or larger ($\delta = 2$) strides lead to performance drops. Incorporating the dynamic stride from V2PE does not shows a significant benefit.

### D.6 FULL EXPERIMENT RESULTS ON QWEN3-VL

We present the complete results of our main experiments on Qwen3-VL. Table 10 and Table 11 report the performance of **Qwen3-VL-4B-Instruct** and **Qwen3-VL-8B-Instruct**, respectively. Across both model scales, MHRoPE and MRoPE-I consistently outperform other multimodal positional encoding variants. The experimental settings are identical to those in the main experiments.

Table 10: Overall performance of multimodal RoPEs variants on various benchmarks, evaluated on Qwen3-VL-4B-Instruct. The highest score is shown in **bold**, while the second-highest score is underlined.

| Types | Benchmarks | Vanilla RoPE | MRoPE | VideoRoPE | HoPE | CircleRoPE | MHRoPE | MRoPE-I |
|-------|-----------|--------------|-------|-----------|------|------------|--------|---------|
| Image | MMMU | 45.78 | 42.89 | 44.44 | 45.44 | 45.33 | 46.11 | **46.33** |
| | MMBench$_{avg}$ | 68.77 | 71.26 | 69.20 | 70.79 | 71.35 | **71.56** | 71.20 |
| | MMstar | 43.67 | 42.87 | 41.87 | 43.53 | 43.73 | 43.73 | **45.00** |
| | OCRBench | 35.20 | 37.30 | 32.10 | 28.70 | 37.10 | **38.60** | 38.30 |
| | AI2D | 66.71 | 67.29 | 63.21 | 63.46 | 66.06 | 68.32 | **69.29** |
| | RealworldQA | 57.12 | 56.60 | 57.25 | 57.52 | 57.25 | 58.86 | **59.78** |
| | DocVQA | 46.59 | 43.13 | 25.27 | 27.25 | 44.60 | 46.77 | **46.89** |
| | TextVQA | 46.65 | 48.25 | 47.10 | 47.16 | 47.27 | **48.60** | 48.27 |
| | InfoVQA | 30.55 | 28.14 | 16.22 | 17.32 | 30.02 | 30.93 | **32.46** |
| | ChartQA | 39.64 | 38.80 | 39.37 | 39.52 | 40.68 | 39.76 | **41.64** |
| | BLINK | 36.33 | 36.80 | 34.44 | 34.46 | 37.52 | 37.22 | **37.88** |
| Video | MVBench | 54.30 | 53.95 | 53.83 | 54.53 | 53.13 | **54.98** | 54.30 |
| | STAR | 58.07 | 57.90 | **58.77** | 56.73 | 55.89 | 58.66 | 58.23 |
| | MLVU | 60.03 | 60.76 | 60.21 | **61.72** | 59.40 | 61.00 | 61.29 |
| | VideoMME | 50.93 | 50.41 | 50.19 | **51.41** | 50.44 | 51.30 | 50.70 |
| | LVBench | 36.93 | 36.35 | 36.09 | 37.02 | 35.54 | **37.20** | 36.99 |
| | Charades-STA | 39.16 | 38.20 | 40.06 | 40.08 | 40.98 | **41.68** | 41.44 |
| Grounding | RefCOCO$_{val}$ | 22.51 | 25.53 | 19.58 | 19.74 | 20.38 | 26.70 | **28.82** |
| | RefCOCO$_{testA}$ | 22.91 | 27.40 | 20.03 | 21.35 | 21.21 | 26.60 | **28.91** |
| | RefCOCO$_{testB}$ | 23.20 | 25.14 | 21.90 | 20.27 | 20.88 | 28.13 | **30.68** |
| | RefCOCO+$_{val}$ | 17.12 | 20.37 | 15.45 | 15.98 | 15.48 | 21.33 | **22.61** |
| | RefCOCO+$_{testA}$ | 18.49 | 21.76 | 15.37 | 16.66 | 16.59 | 21.87 | **23.85** |
| | RefCOCO+$_{testB}$ | 18.20 | 20.37 | 17.71 | 16.32 | 16.22 | 23.54 | **25.81** |
| | RefCOCOg$_{val}$ | 22.32 | 24.90 | 20.34 | 20.45 | 20.10 | 26.80 | **30.00** |
| | RefCOCOg$_{test}$ | 21.38 | 23.89 | 20.93 | 19.19 | 19.82 | 26.90 | **29.52** |
| Overall | Image | 47.00 | 46.67 | 42.77 | 43.20 | 47.36 | 48.22 | **48.82** |
| | Video | 49.90 | 49.60 | 49.86 | 50.25 | 49.23 | **50.80** | 50.49 |
| | Grounding | 20.77 | 23.67 | 18.91 | 18.74 | 18.84 | 25.23 | **27.52** |

Table 11: Overall performance of multimodal RoPEs variants on various benchmarks, evaluated on Qwen3-VL-8B-Instruct. The highest score is shown in **bold**, while the second-highest score is underlined.

| Types | Benchmarks | Vanilla RoPE | MRoPE | VideoRoPE | HoPE | CircleRoPE | MHRoPE | MRoPE-I |
|---|---|---|---|---|---|---|---|---|
| Image | MMMU | 51.18 | 50.11 | 51.78 | 51.89 | 50.67 | 53.33 | **53.89** |
| | MMBench$_{avg}$ | 79.29 | 78.27 | 78.74 | 79.59 | 79.00 | **80.78** | 79.50 |
| | MMstar | 51.88 | 52.53 | 53.33 | 52.86 | **54.40** | 53.20 | 53.47 |
| | OCRBench | 73.20 | 72.90 | 67.70 | 61.70 | 73.60 | **74.40** | 73.90 |
| | AI2D | 78.90 | 77.56 | 78.34 | 77.85 | 77.59 | 78.22 | **79.50** |
| | RealworldQA | 63.27 | 64.05 | 62.48 | 62.48 | 61.96 | **67.33** | 65.22 |
| | DocVQA | 82.41 | 82.85 | 71.71 | 72.95 | 82.28 | 82.41 | **83.67** |
| | TextVQA | 63.15 | 66.05 | 61.54 | 62.33 | 64.94 | 67.20 | **67.32** |
| | InfoVQA | **53.84** | 49.60 | 39.15 | 41.69 | 51.06 | 52.22 | 52.62 |
| | ChartQA | 62.52 | 61.04 | 61.00 | 59.04 | 62.96 | 61.89 | **63.44** |
| | BLINK | 37.21 | 37.44 | 37.08 | 37.22 | 40.52 | 40.00 | **40.45** |
| Video | MVBench | 60.35 | 59.90 | 59.70 | 60.53 | 59.53 | 60.20 | **60.70** |
| | STAR | 61.65 | 62.29 | 61.21 | 62.19 | 61.51 | 62.33 | **62.45** |
| | MLVU | 66.01 | 67.34 | 68.31 | **68.81** | 66.61 | 67.20 | 67.52 |
| | VideoMME | 60.07 | 59.78 | **61.48** | 60.56 | 51.04 | 60.89 | 60.89 |
| | LVBench | 42.22 | 41.06 | **42.93** | 42.48 | 38.80 | 42.00 | 41.58 |
| | Charades-STA | 50.39 | 51.79 | **52.90** | 52.30 | 51.77 | 52.13 | 52.70 |
| Grounding | RefCOCO$_{val}$ | 73.22 | 74.16 | 71.68 | 73.22 | 74.70 | 76.88 | **77.72** |
| | RefCOCO$_{testA}$ | 76.45 | 77.11 | 73.77 | 74.50 | 76.77 | 79.41 | **83.21** |
| | RefCOCO$_{testB}$ | 71.40 | 72.09 | 69.89 | 70.93 | 73.22 | 72.89 | **76.45** |
| | RefCOCO+$_{val}$ | 65.77 | 65.94 | 62.05 | 66.33 | 66.37 | 67.99 | **70.88** |
| | RefCOCO+$_{testA}$ | 71.17 | 72.25 | 68.08 | 72.89 | 72.41 | 75.08 | **77.79** |
| | RefCOCO+$_{testB}$ | 59.97 | 60.46 | 59.26 | 60.58 | 61.16 | 63.18 | **64.33** |
| | RefCOCOg$_{val}$ | 71.14 | 73.10 | 69.77 | 71.28 | 72.33 | 74.14 | **76.47** |
| | RefCOCOg$_{test}$ | 71.57 | 72.68 | 69.45 | 71.82 | 72.67 | 74.65 | **76.79** |
| Overall | Image | 63.35 | 62.95 | 60.26 | 59.96 | 63.54 | 64.63 | **64.82** |
| | Video | 56.78 | 57.03 | 57.75 | **57.81** | 54.88 | 57.46 | 57.64 |
| | Grounding | 70.09 | 70.97 | 67.99 | 70.19 | 71.20 | 73.03 | **75.46** |

