# OpenReview forum: "Revisiting Multimodal Positional Encoding in Vision–Language Models"
_ICLR.cc/2026/Conference — ICLR 2026 Poster_

### Official Review · Reviewer_kJVi · 2025-10-28

**Soundness:** 3
**Presentation:** 3
**Contribution:** 3
**Rating:** 6
**Confidence:** 4

**Summary:**

This paper focuses on the design of multimodal position embeddings for VLMs. Specifically, the authors analyze existing RoPE variants and their designs used in VLMs, and summarize three key guidelines: the position design for different modalities, the frequency allocation for different positional axes, and the compatibility with vanilla text-only RoPE. Guided by these principles, the paper proposes two new designs: MHRoPE and MRoPE-I. Both methods enhance the MRoPE by incorporating a ‘spatial-reset' mechanism, and they introduce two novel frequency allocation strategies. Experiments demonstrate the effectiveness of the two proposed methods.

**Strengths:**

* The paper provides a comprehensive and systematic discussion of the multimodal RoPE methods.
* The proposed two methods MHRoPE and MRoPE-I are simple while well-motivated, and the results show their effectiveness.
* There are extensive experiments, with various benchmarks across image understanding, video understanding and visual grounding tasks.

**Weaknesses:**

* Validation on only a single VLM instance (i.e. the experiments are based on Qwen2.5VL) may not be sufficient to demonstrate the robustness of the methods.
* It is surprising that MRoPE underperforms Vanilla RoPE on many benchmarks (Table 2). Perhaps the authors could provide a deeper discussion about the results.

**Questions:**

For MHRoPE, is there any experiment or discussion on how the heads are allocated?

---

> ### Author Response · Authors · 2025-11-25
>
> We thank the reviewer for the concise and constructive feedback. We have addressed your concerns point-by-point below.
>
> > W1: Validation on only a single VLM instance (i.e. the experiments are based on Qwen2.5VL) may not be sufficient to demonstrate the robustness of the methods.
>
> **A1**: We acknowledge that relying solely on Qwen2.5-VL was a limitation. To demonstrate the robustness of our methods, we have expanded our validation to the **Qwen3-VL family (4B and 8B)**, which introduces major architectural shifts such as QK-Norm and a multi-scale DeepStack connector.
> Please refer to **General Response Point 1** for the detailed results. The consistent improvements observed on Qwen3-VL confirm that the effectiveness of MHRoPE and MRoPE-I is robust and not limited to a single VLM instance.
>
> > W2: It is surprising that MRoPE underperforms Vanilla RoPE on many benchmarks (Table 2). Perhaps the authors could provide a deeper discussion about the results.
>
> **A2**: This result is actually expected. Although MRoPE introduces extra spatio-temporal information via its position design, it suffers from severe **uneven frequency allocation**, which damages its capabilities in long-video modeling (as seen in VideoRoPE/HoPE) and fine-grained spatial understanding (as seen in Mogao). In contrast, while Vanilla RoPE does not carry explicit spatial information, it retains a complete frequency allocation without introducing position ID ambiguity, allowing it to perform on par with MRoPE on standard benchmarks. However, Vanilla RoPE has a critical flaw: its position IDs grow too rapidly. As shown in **Appendix D.5 (Figure 5)**, Vanilla RoPE degrades significantly in long-sequence extrapolation (e.g., training on 32K and testing on 64/128/256K). This is unacceptable given the increasing demand for long-context VLMs.
>
> > Q1: For MHRoPE, is there any experiment or discussion on how the heads are allocated?
>
> **A1**: In our implementation of MHRoPE, the allocation of heads across T, H, and W dimensions is random. We experimented with different allocation strategies, but they did not show significant performance differences. Although different heads in an LLM may have distinct attention patterns, we found that after fine-tuning, the heads assigned to spatial understanding (H and W) can adapt well to their roles.
> There is, however, an important detail mentioned in Footnote 1 of our paper regarding Group Query Attention (GQA). Since one Key/Value head interacts with multiple Query heads, we must avoid mixing positional signals within a group. Therefore, we enforce a constraint where all Query heads corresponding to the same Key/Value head pair are allocated the same type of position information (e.g., all assigned to T).

---

### Official Review · Reviewer_T4pq · 2025-10-30

**Soundness:** 3
**Presentation:** 3
**Contribution:** 2
**Rating:** 6
**Confidence:** 3

**Summary:**

This paper presents a systematic study of multimodal rotary positional embeddings (RoPE) in Vision-Language Models (VLMs). The authors decompose the multimodal RoPE design space into three core aspects—position design, frequency allocation, and compatibility with text-only RoPE—and identify common pitfalls in existing methods. Building upon these insights, they propose two new variants: Multi-Head RoPE (MHRoPE) and MRoPE-Interleave (MRoPE-I), along with a spatial-reset mechanism. These designs aim to ensure positional coherence, full frequency utilization, and faithful transfer of textual priors from pre-trained LLMs. Through extensive experiments on over twenty benchmarks covering image, video, and grounding tasks, the proposed methods achieve consistent and notable improvements over prior work.

**Strengths:**

- The paper provides one of the most systematic and insightful examinations of positional encoding in multimodal contexts to date. The decomposition into design axes (position, frequency, and compatibility) clarifies the often opaque design space of VLM encodings.

- MHRoPE and MRoPE-I are conceptually straightforward and “plug-and-play,” requiring no architectural modifications, which increases their practical utility.

- The authors evaluate across a wide range of benchmarks, demonstrating consistent gains and analyzing failure cases of prior methods.

- Extensive ablations provide convincing evidence for the effectiveness of spatial-reset and interleaved frequency allocation.

**Weaknesses:**

- While the analysis is systematic, the theoretical understanding of why certain frequency allocations or resets perform better remains mostly empirical. A deeper mathematical treatment could strengthen the claims.
- Although improvements are consistent, in some cases (e.g., video benchmarks), the gains are modest, which might question the generality of the benefits.
- Dependence on Qwen2.5-VL backbone. All experiments rely on one model family. It is unclear if the conclusions would hold equally well for other architectures (e.g., LLaVA or Gemini-style models).
- The authors note MHRoPE’s potential scalability but provide little empirical evidence of its performance on extremely large-scale or high-resolution data.
- The improvement in the experimental results is not significant.

**Questions:**

- Carefully review Table 2. It seems that the calculations for some data are incorrect, especially in the "overall" column and the parts in red font.
- How sensitive are the proposed methods to the choice of rotary base or context length during training? Would a smaller base or shorter context alter the observed trends?
- Can the authors elaborate on whether spatial-reset interacts with patch embedding granularity (e.g., ViT patch size)?
- Is there a possibility that interleaved frequency allocation could interfere with frequency-aware extrapolation methods when used in ultra-long video sequences?

---

> ### Author Response · Authors · 2025-11-25
>
> We sincerely thank the reviewer for the systematic review and insightful questions. We have carefully addressed each point below.
>
> > W1: Theoretical understanding vs. empirical evidence (Frequency allocations & Resets).
>
> **A1**: Regarding Frequency Allocation, we provide a mathematical derivation in Appendix D.2 (illustrated in Figure 4) that explains how uneven frequency allocation leads to long-term decay. This serves as the theoretical basis for the superiority of our interleaved strategy.
>
> Regarding spatial-reset, our design is motivated by the empirical observation of "attention sinks" in visual/video tokens. While a unified theoretical explanation for attention sinks in LLMs remains an active research topic, our ablation studies (Table 3) and quantitative analysis of visual attention scores (Appendix D.5.1) provide strong empirical validation.
>
> > W2: Modest gains in video / Generality of benefits.
>
> **A2**: Specialized methods like VideoRoPE and HoPE optimize aggressively for long-video understanding (e.g., by allocating all low-frequency channels to temporal positions), but this comes at a steep cost: significant degradation in high-resolution document understanding and grounding (see Table 2).
> In contrast, our method prioritizes **universal VLM capabilities**. We achieve the best comprehensive performance across image, video, and grounding tasks without sacrificing one for the other. Furthermore, our work contributes by systematically revisiting multimodal RoPEs and summarizing three key design insights, providing valuable guidelines for the design of future general-purpose VLMs.
>
> > W3: Dependence on Qwen2.5-VL backbone.
>
> **A3**: We thank the reviewer for raising this valid concern. Regarding Gemini, since its architecture is proprietary, we cannot modify its internal positional encodings. Regarding LLaVA, its architecture is highly similar to Qwen2.5-VL. To rigorously test robustness on a distinct architecture, we selected the newly released **Qwen3-VL**, which differs significantly from the standard designs (e.g., tied embeddings in 4B, DeepStack connector and QK-Norm). As shown in the new experimental results in the **General Response point 1**, our conclusions hold equally well for this new architecture, confirming that our method is not dependent on the specific Qwen2.5 backbone.
>
> > W4: Lack of empirical evidence for MHRoPE’s scalability.
>
> **A4**: We would like to clarify that **MHRoPE's "scalability" refers to dimensional scalability rather than just data scaling**. Unlike existing methods that must carve up the frequency spectrum within a single attention head, MHRoPE leverages the redundancy across different attention heads. This allows it to scale efficiently to higher-dimensional inputs (e.g., adding 3D coordinates (x,y,z) or additional modalities) without diluting the frequency capacity assigned to each dimension.
>
> > W5: The improvement is not significant.
>
> **A5**: We respectfully disagree. According to Table 2, MHRoPE and MRoPE-I achieve average performance gains of **1.89%**, **0.85%**, and **2.16%** over MRoPE across 20+ benchmarks. These gains are even more pronounced on complex tasks, such as **BLINK (+6.15%)** and **InfoVQA (+5.28%)**. Considering that these improvements are achieved solely through an improved PE strategy **without altering the model architecture or adding inference cost**, we believe the contribution is substantial and practically valuable.

---

> > ### Author Response · Authors · 2025-11-25
> >
> > > Q1: Calculation errors in Table 2.
> >
> > **A1**: Thank you for your meticulous check! We have carefully reviewed and corrected the scores in Table 2. The "Overall" score is calculated as the simple arithmetic mean of the benchmark categories.
> >
> > > Q2: Sensitivity to rotary base or context length.
> >
> > **A2**: Empirically, the choice of rotary base is strongly correlated with context length (larger bases are preferred for longer contexts). We conducted an ablation at **32k context length** comparing **base=1M** vs. **base=500k**. As shown below, while Base=1M generally performs better, the relative performance advantage of MRoPE-I over MRoPE holds consistent regardless of the base.
> >
> > | Method | Image | Video | Grounding |
> > | :--- | :---: | :---: | :---: |
> > | MRoPE (1M) | 61.90 | 51.51 | 73.69 |
> > | MRoPE (500k) | 61.32 | 49.13 | 73.50 |
> > | **MRoPE-I (1M)** | **63.79** | **52.36** | **75.85** |
> > | MRoPE-I (500k) | 63.27 | 51.00 | 75.22 |
> >
> > > Q3: Interaction between spatial-reset and patch embedding granularity.
> >
> > **A3**: Since the attention sink phenomenon typically occurs at the token level, we hypothesized that patch size would not fundamentally alter the trend. While we cannot retrain a ViT with varying patch sizes from scratch, our new experiments on Qwen3VL (Patch Size 16) vs. Qwen2.5-VL (Patch Size 14) provide a proxy comparison. MHRoPE/MRoPE-I remains superior in both cases, indirectly validating that the effectiveness of spatial-reset is robust to patch granularity.
> >
> > > Q4: Interference with frequency-aware extrapolation methods (e.g., YaRN).
> >
> > **A4**: We discuss this compatibility in Appendix D.3. To empirically verify it, we tested MRoPE/MRoPE-I combined with YaRN directly in a 256K context scenario. As shown below (also added to Appendix Table 6), MRoPE-I actually benefits more from YaRN than the baseline does (+1.6 vs -0.3 on LVBench), demonstrating superior extrapolation potential.
> >
> > | Method | LVBench | MLVU |
> > | :--- | :---: | :---: |
> > | MRoPE | 41.5 | 62.9 |
> > | MRoPE + YaRN | 41.2 (-0.3) | 63.3 (+0.4) |
> > | MRoPE-I | 42.0 | 63.2 |
> > | **MRoPE-I + YaRN** | **43.6 (+1.6)** | **64.1 (+0.9)** |

---

### Official Review · Reviewer_PFUK · 2025-10-31

**Soundness:** 2
**Presentation:** 3
**Contribution:** 2
**Rating:** 4
**Confidence:** 3

**Summary:**

The paper proposes MRoPE-I and MHRoPE, two improved methods for using rotary embeddings to encode position information in VLMs. MRoPE-I and MHRoPE improve on existing techniques by:introducing a "spatial-reset" to fully decouple position and time encoding and ensuring that positional infomration is encoded across the entire frequency spectrum. The work proposes with two methods for encoding frequency across the full spectrum: head specialization (MHRoPE) and interleaved (MRoPe-I).

**Strengths:**

- In the reivewers opinion, the ideal way to encode position in VLMs is unclear and warrants further exploration. In particular, well put together ablations are interesting and useful.

- The severe degradation of vanilla RoPE on long video inputs (and relatively strong performance on shorter visual tasks) and observation of a “visual attention sink“ across modalities, is interesting.

- The paper is clear in its presentation of current challenges in multimodal position embeddings, and its proposed solution.

**Weaknesses:**

- The vision encoder of Qwen2VL / connector MLP of Qwen2.5VL has been trained to output visual token that are used with MRoPE-like position encodings. This may disadvantage approach such as VideoRoPE, HoPE and CircleRoPE. Initializing from  vision encoder that has not been trained for use in a VLM would be better. This problem also seems to result in disabling the absolute time encoding scheme proposed for MRoPe [1]. Therefore, the MRoPe baseline is not one-to-one with how it is used in QwenVL-2.5. Overall, heavy use of QwenVL components makes interpreting experiments difficult and introduces confounding factors.


- The experimental setup is quite limited. Only 1 LLM is considered, and only at 7B parameter size. Showing generalization across architectures would be valuable.


- The stages followed for training the VLMs are not standard, especially the choices to initialize the vision encoder and MLP layers with QwenVL components. It would be better to recreate an established process for constructing a VLM from existing work [2,3].


- A discussion about the similarities of MRoPE-I and MHRoPE to MRoPE **given the training and testing distribution** is essential. In the case of a single visual element and no system prompt the proposed approach is very similar MRoPE. What/if system prompts are used for each evaluation is not clear. However, the single image case is certainly the dominant case among evaluations. Are system prompts are used during training? Is the system prompt of consistent length during training / evaluation? This is important for understanding exactly how position is encoded (i.e. what do equations 3 and 4 realistically evaluate to).


- The paper is very vague about what training data is used. The reviewer can only find that “2M high-quality supervised fine-tuning (SFT) samples” were used. Nothing about the breakdown between modalities, conversations vs. captioning, particular data sources, etc. Figure 1 demonstrates the position encoding for conversations containing multiple interleaved visual elements. Did the training contain conversations with multiple visual elements, if so, what proportion of the training data was it? Such details are crucial for understanding how position is realistically encoded and reproducibility.


- Some details are missing from training settings, such as the optimizer used and it’s respective settings. Justification for the choices of hyperparameters would also be valuable.


- It is claimed that MHRoPE / MRoPE-I result in “more focus on visual information.“ However, that is never justified experimentally or formalized. Do the proposed methods have larger attention scores (on visual tokens) than baselines?


- The benchmarks used to evaluate MRoPE-I and MHRoPE overwhelmingly follow the structure of a single visual document followed by some text (usually a short question). This does not test the proposed method across a diverse range of positional encoding setups. Comparisons across conversations with multiple visual documents (akin to Figure 1) would be valuable.

Minor:

- Table 2 is a bit messy. Sometimes the best score is bold and in other rows the second best score is bold (for example, the Video row).

- Quotes open in the wrong direction in the manuscript, and are inconsistent throughout. Some are correct such as “1111...” whereas elsewhere it is flipped, such as ”attention sink," .

**Questions:**

Several question can be found in the section above, in addition:

- What is the performance if the number of temporal channels is less than 24 in the interleaved strategy?

- How are video frames sampled? Are a constant number of frames sampled uniformly or is fps sampling used? If so, what fps is used? Similarly, what image processing/scaling is used? Do the authors use QwenVL2.5’s dynamics scaling of visual tokens? Are the total number of frames sampled capped in long video? Do any of these settings differ between training and evaluation?

- How was the ChartQA visualization created in figure 2? It is the reviewer’s understanding that ChartQA has a single image per input, but figure 2 has two images.

---

[1] S. Bai *et al.*, “Qwen2.5-VL Technical Report,” 2025, doi: 10.48550/arxiv.2502.13923.

[2] O. Zohar *et al.*, “Apollo: An Exploration of Video Understanding in Large Multimodal Models,” in *Proceedings (IEEE Computer Society Conference on Computer Vision and Pattern Recognition. Online)*, IEEE, 2025, pp. 18891–18901. doi: 10.1109/CVPR52734.2025.01760.

[3] S. Tong *et al.*, “Cambrian-1: A Fully Open, Vision-Centric Exploration of Multimodal LLMs,” 2024, doi: 10.48550/arxiv.2406.16860.

---

> ### Author Response · Authors · 2025-11-25
>
> We thank the reviewer for the insightful comments and detail feedback. We have carefully addressed each point below.
>
> > W1 & W3: Heavy use of QwenVL components introduces confounding factors / Non-standard training stages.
>
> **A1 & A3**: First, it is important to clarify that **the vision encoder of Qwen2-VL utilizes standard 2D RoPE**, distinct from the MRoPE used in the LLM decoder; thus, it is not inherently biased towards MRoPE-like logic. Second, our study aims to isolate **the impact of multimodal positional encodings within the LLM**. By freezing the vision encoder and connector while re-initializing the LLM backbone, we establish a controlled environment where the PE strategy is the sole variable—a protocol also followed by baselines like VideoRoPE and HoPE. While works like Apollo and Cambrian-1 explore general VLM architectural pipelines (training connector from scratch), adopting such an approach here would introduce unnecessary confounding factors (e.g., alignment instability). We have updated **Section 3.1** to explicitly clarify this scope and justification.
>
> > W2: The experimental setup is quite limited (only 7B). Showing generalization across architectures would be valuable.
>
> **A2**: We agree that validating generalization across architectures and scales is crucial. To address this, we performed additional experiments on **Qwen3-VL-4B-Instruct** and **Qwen3-VL-8B-Instruct**.
> As discussed in **General Response Point 1**, Qwen3-VL represents a structurally distinct architecture compared to Qwen2.5-VL (e.g., employing tied embeddings in 4B, removing window attention in ViT, and using a DeepStack connector). Please refer to the Table in General Response Point 1 and Section 3.3, where our results demonstrate that MHRoPE and MRoPE-I maintain their superiority across different parameter scales (4B/8B) and architectural paradigms.
>
> > W4: Similarity to MRoPE in single-image cases / Role of system prompts.
>
> **A4**:
> 1. System Prompts: Consistent with Qwen2-VL and VLMEvalKit standards, system prompts are always prepended before images. During training, prompts vary; during evaluation, we fix the prompt to "You are a helpful assistant."
> 2. Impact of Spatial-Reset: Evidence from multi-image benchmarks (e.g., MMMU, BLINK) and our ablation study (Table 3) demonstrates that MHRoPE and MRoPE-I benifit from spatial-reset.
> 3. Eq. 3 & 4: These equations quantify the spatial-reset effect on video consistency. By resetting positions, we ensure better spatio-temporal alignment across frames, which directly contributes to the performance gains observed in video benchmarks compared to MRoPE (Table 3).
>
> > W5: Vague training data description.
>
> **A5**: We utilized in-house supervised fine-tuning (SFT) data, covering diverse tasks including VQA, OCR, pure text, and **multi-turn dialogue**. While we cannot disclose the exact proprietary data mixture ratios, we are committed to open-sourcing our code and model checkpoints to ensure reproducibility of our results.
>
> > W6: Missing training details (optimizer, hyperparameters).
>
> **A6**: Thank you for pointing this out. We used the AdamW optimizer (α= 0.9 and β=0.98) with a weight decay of 0.05. These details have been added to Section 3.1 (Experiment Setup).
>
> > W7: Claim of "more focus on visual information" is not justified experimentally.
>
> **A7**: We performed a quantitative analysis of attention scores in Appendix D.5.1. We measured the visual attention scores (normalized againstt the multimodal sequence) across layers. The results confirm that with spatial-reset, the model assigns higher attention scores to visual information in deeper layers compared to the baselines.
>
> > W8: Benchmarks overwhelmingly follow single-image structure.
>
> **A8**: We acknowledge the importance of multi-image evaluation. Beyond the multi-image samples already present in MMMU, we have now evaluated our method on the **BLINK** benchmark, which specifically targets complex interleaved image tasks. As shown below, MHRoPE (**42.80**) and MRoPE-I (**44.08**) significantly outperform MRoPE (37.93) and other baselines, demonstrating superior robustness in multi-image scenarios.
>
> | | Vanilla RoPE | MRoPE | VideoRoPE | HoPE | CircleRoPE | MHRoPE | MRoPE-I |
> | :--- | :---: | :---: | :---: | :---: | :---: | :---: | :---: |
> | **BLINK** | 36.12 | 37.93 | 36.20 | 35.98 | 40.88 | 42.80 | 44.08 |

---

> > ### Author Response · Authors · 2025-11-25
> >
> > > Minor: Table 2 formatting and quote direction.
> >
> > **Response**: Thank you for your detailed review. We have corrected the formatting in Table 2 and fixed the inconsistent quote marks throughout the manuscript.
> >
> > > Q1: What is the performance if the number of temporal channels is less than 24?
> >
> > **A1**: We have completed an ablation study on frequency allocation ratios (T:H:W). As shown in the table below, completely ignoring temporal information (T=0) significantly degrades video performance. Reducing temporal channels (T=12) results in a loss of high-frequency temporal details, slightly hurting video tasks. A **balanced distribution (24:20:20)** ensures each axis captures a sufficient frequency range, yielding the best overall performance across image, video, and grounding tasks.
> >
> > | Allocation Ratio | Image | Video | Grounding | Overall |
> > | :---: | :---: | :---: | :---: | :---: |
> > | 0:32:32 | 66.42 | 51.01 | 76.02 | 64.48 |
> > | 12:26:26 | 66.30 | 51.93 | 75.77 | 64.67 |
> > | 24:20:20 | 66.65 | 52.36 | 75.85 | **64.95** |
> > | 32:16:16 | 64.07 | 51.15 | 74.65 | 63.29 |
> > | 48: 8: 8 | 65.06 | 51.17 | 72.87 | 63.03 |
> >
> > > Q2: Is dynamic resolution used? How are video frames sampled?
> >
> > **A2**: We follow the standard Qwen2-VL protocol for both training and evaluation. Specifically, we utilize **dynamic resolution scaling** for all visual inputs. For video sampling, we generally use **2 FPS**. In long-video scenarios, we switch to uniform sampling capped at **768 frames** (extended to **2048 frames** for our long-video extrapolation experiments). These evaluation details have been explicitly clarified in **Section 3.1 Evaluation Setup**.
> >
> > > Q3: How was the ChartQA visualization created in Figure 2? (It has two images).
> >
> > **A3**: Thank you for the keen observation. This visualization was designed as a probing experiment to investigate the "attention sink" phenomenon. We deliberately repeated the ChartQA image using the prompt structure: "Image 1: <img> Image 2:  <img> Describe the two images in detail." This allows us to visualize whether the model maintains attention on both instances or succumbs to a sink. We have clarified this experimental design in the paper.

---

### Official Review · Reviewer_KSo7 · 2025-11-03

**Soundness:** 3
**Presentation:** 3
**Contribution:** 3
**Rating:** 4
**Confidence:** 3

**Summary:**

The paper re-examines how multimodal positional encodings (especially RoPE) should be designed for vision–language models, arguing that current extensions are ad hoc and often break either visual geometry or LLM compatibility. It identifies three principles—positional coherence, full frequency utilization, and preservation of textual priors—as necessary to avoid modality confusion, attention sinks, and degraded long-context/video performance. Based on this, the authors propose two plug-and-play variants, Multi-Head RoPE (MHRoPE) and MRoPE-Interleave (MRoPE-I), which give each axis (time, height, width) access to the full frequency spectrum while keeping text RoPE unchanged. Extensive experiments on >20 image, video, and grounding benchmarks show these designs consistently outperform vanilla RoPE, MRoPE, and recent video-focused RoPEs, especially on fine-grained and grounding tasks.

**Strengths:**

1. propose two plug-and-play variants, Multi-Head RoPE (MHRoPE) and MRoPE-Interleave (MRoPE-I), which give each axis (time, height, width) access to the full frequency spectrum while keeping text RoPE unchanged.
2. Extensive experiments on >20 image, video, and grounding benchmarks prove effectiveness of these designs
3. Easy to understand and comprehend. Figures are clear.

**Weaknesses:**

1. Limited experimented VLMs. The experiments only cover Qwenvl2.5. Broader range of models such as llama, phi and internvl series would better demonstrates the generalizability of the benefits from the two novel designs.
2. Increamental technical novelty compared with MRoPE. This paper propose spatial-reset and two frequency allocation modifications upon MRoPE. Despite effectiveness, the overall technical novelty compared to MRoPE and the original RoPE remains limited.
3. experiment with rapid temporal decay and spatial asymmetry problem, such as thousands of frames or extreme interleaving patterns where attention sinks typically reappear and see the performance

**Questions:**

Is MHRoPE and MRoPE-I more scalable than MRoPE and RoPE in the multimodal domain, such as better loss curve with proportionally increasing model size and data size?

---

> ### Author Response · Authors · 2025-11-25
>
> We thank the reviewer for the insightful comments and constructive feedback. We have carefully addressed each point below.
>
> > W1: Limited experimented VLMs. The experiments only cover Qwen2.5VL.
>
> **A1:** We appreciate the constructive suggestion to broaden the experimental scope. As detailed in **General Response Point 1**, we have conducted comprehensive new experiments on the **Qwen3-VL** series (4B and 8B).
> We chose Qwen3-VL because models like Llama 4 and Phi-3.5-Vision share a highly homogenous architecture with Qwen2.5-VL (Standard ViT + MLP), whereas Qwen3-VL introduces significant structural deviations (DeepStack connector, QK-Norm, tie-embeddings). As shown in the Table in General Response Point 1, MHRoPE and MRoPE-I consistently outperform baselines on Qwen3-VL, confirming that the benefits of our design are generalizable across different architectures.
>
> > W2: Incremental technical novelty compared with MRoPE.
>
> **A2:** While we acknowledge the pioneering contributions of the original RoPE and MRoPE to the field, they are not universally optimal. Specifically, MRoPE and its follow-ups exhibit inherent design limitations regarding frequency allocation and ID design, which we analyze in depth. **The core technical novelty of our work lies in systematically identifying these deficiencies and proposing a theoretically grounded solution.** We believe that rectifying these fundamental issues provides a necessary cornerstone for the design of general vision-language models, offering value beyond incremental improvements.
>
> > W3: Experiment with rapid temporal decay and spatial asymmetry problem.
>
> **A3:** Thank you for raising these points regarding extreme scenarios.
> 1. Rapid Temporal Decay: We address long-context performance in Appendix D.4. By extending the context to 256K (approx. 2048 frames), we observed that while Vanilla RoPE suffers a sharp performance drop, MRoPE-based methods (including ours) demonstrate robust video extrapolation capabilities.
> 2. Spatial Asymmetry: We investigate this in Section 3.3.1 and Table 3 by ablating the diagonal layouts used in VideoRoPE and HoPE. Our results show that enforcing spatial symmetry leads to model confusion on high-resolution inputs (e.g., DocVQA, InfoVQA). This is further corroborated by our new experiments on the Qwen3VL 4B/8B architectures (see Sec 3.1 or Appendix D.6).
> 3. Attention Sink in Extreme Interleaving: In massive interleaving patterns, the attention sink phenomenon is diluted across numerous images and is hard to observe. However, to demonstrate effectiveness in multi-image settings, we evaluated on the BLINK benchmark. MHRoPE (**42.8**) and MRoPE-I (**44.08**) significantly outperform MRoPE (37.93), proving our method handles complex image sequences more effectively.
>
> | | Vanilla RoPE | MRoPE | VideoRoPE | HoPE | CircleRoPE | MHRoPE | MRoPE-I |
> | :--- | :---: | :---: | :---: | :---: | :---: | :---: | :---: |
> | **BLINK** | 36.12 | 37.93 | 36.20 | 35.98 | 40.88 | 42.80 | 44.08 |
>
> > Q1: Is MHRoPE and MRoPE-I more scalable than MRoPE and RoPE in the multimodal domain (e.g., better loss curve)?
>
> **A1:** In our experiments, **Vanilla RoPE shows a lower training loss** simply because it aligns with the LLM's pre-trained 1D RoPE prior, requiring minimal adaptation. In contrast, MRoPE-based methods introduce new 3D spatial logic, naturally incurring a higher initial adaptation loss. Crucially, however, this higher loss does not impede capability; MRoPE-based methods achieve superior performance on downstream tasks. Therefore, we rely on **broad benchmark performance** rather than the training loss curve as the true indicator of scalability and effectiveness in the multimodal domain.

---

### Author Response · Authors · 2025-11-25
**General Response to All Reviewers**

We thank all reviewers (KSo7, PFUK, T4pq, kJVi) for their time, insightful comments, and constructive suggestions. We are encouraged that the reviewers found our analysis systematic and our proposed methods (MHRoPE and MRoPE-I) effective.

In response to the feedback, we have strengthened the manuscript by **expanding the experimental scope to a structurally distinct VLM architecture (Qwen3-VL)**, **evaluating on complex multi-image benchmarks (BLINK)**, and **verifying MRoPE-I’s compatibility with long-sequence video extrapolation (YaRN)**. Below is a summary of the major updates and clarifications.

---

**1. Generalization Across Architectures: Experiments on Qwen3-VL (Response to KSo7 W1, PFUK W2, T4pq W3, kJVi W1)**

To address concerns regarding the reliance on the Qwen2.5-VL family, we expanded our evaluation. We note that directly modifying proprietary models (e.g., Gemini) is infeasible, while most other open-source VLMs (e.g., Llam4, Phi-3.5-Vision, InternVL3) share highly homogenous architectures with Qwen2.5 (Standard ViT + MLP + Standard LLM), offering limited value in testing architectural robustness. Instead, we selected the newly released Qwen3-VL as a stricter testbed. Unlike standard paradigms, Qwen3-VL introduces some structural changes, including **Siglip2-like vision encoder (remove ViT window attention)**, **QK-Norm (LLM part)**, **tied embeddings (4B variant)**, and **a multi-scale DeepStack connector**. As shown below, MHRoPE and MRoPE-I consistently outperform baselines on Qwen3-VL (4B/8B), confirming that our method works effectively across distinct model architectures and is not overfitted to specific configurations. (full result posted on Sec 3.3 and Appendix D.6)


| Qwen3-VL-8B-Instruct | RoPE | MRoPE | VideoRoPE | HoPE | CircleRoPE | **MHRoPE** | **MRoPE-I** |
| :--- | :---: | :---: | :---: | :---: | :---: | :---: | :---: |
| **Image** | 63.35 | 62.95 | 60.26 | 59.96 | 63.54 | 64.63 | **64.82** |
| **Video** | 56.78 | 57.03 | 57.75 | **57.81** | 54.88 | 57.46 | 57.64 |
| **Grnd** | 70.09 | 70.97 | 67.99 | 70.19 | 71.20 | 73.03 | **75.46** |

| Qwen3-VL-4B-Instruct | RoPE | MRoPE | VideoRoPE | HoPE | CircleRoPE | **MHRoPE** | **MRoPE-I** |
| :--- | :---: | :---: | :---: | :---: | :---: | :---: | :---: |
| **Image** | 47.00 | 46.67 | 42.77 | 43.20 | 47.36 | 48.22 | **48.82** |
| **Video** | 49.90 | 49.60 | 49.86 | 50.25 | 49.23 | **50.80** | 50.49 |
| **Grnd** | 20.77 | 23.67 | 18.91 | 18.74 | 18.84 | 25.23 | **27.52** |

**2. Robustness in Complex Scenarios: Multi-Image & Long-Context (Response to KSo7 W3, PFUKv W7, T4pq Q2/4)**

To verify performance in extreme scenarios, we conducted additional evaluations on interleaved images and ultra-long video contexts:
*   **Interleaved Images (BLINK Benchmark):** We evaluated on BLINK [1], a challenging benchmark for multi-image perception. MHRoPE (**42.80**) and MRoPE-I (**44.08**) significantly outperform MRoPE (37.93) and Vanilla RoPE (36.12), proving that our spatial-reset strategy effectively mitigates attention sinks in multi-image sequences.
*   **Long-Video & Extrapolation Compatibility:** We extended the evaluation context to 256K (approx. 2048 frames) in Appendix D.4. Results show that MRoPE-I maintains robust performance while Vanilla RoPE collapses. Furthermore, we verified the compatibility of our method with frequency-aware extrapolation techniques (YaRN). As detailed in **Appendix D.3**, combining MRoPE-I with YaRN yields further gains (+1.6% on LVBench) over the baseline, confirming that our interleaved frequency allocation is compatible with existing extrapolation methods.

**3. Implementation Details & Reproducibility (Response to PFUK W4/5/6)**

We have updated the manuscript to include missing details requested by Reviewer PFUK, including optimizer settings, data composition strategies, and clear definitions of the training/evaluation protocol (**Sec 3.1**).

---

We have uploaded the revised PDF, where **all modifications are highlighted in blue**. We hope these additional experiments and clarifications address the reviewers' concerns. We remain open to further discussions.


**References**

[1] Fu, X., Hu, Y., Li, B., Feng, Y., Wang, H., Lin, X., Roth, D., Smith, N. A., Ma, W.-C., & Krishna, R. (2024). BLINK: Multimodal large language models can see but not perceive. In *Proceedings of the European Conference on Computer Vision (ECCV)*. https://doi.org/10.1007/978-3-031-73337-6_9

---

### Meta-Review · Area_Chair_8mH3 · 2026-01-06

**Summary:**

The paper presents a systematic analysis of Multimodal Rotary Positional Embeddings (RoPE) in Vision-Language Models (VLMs). The authors identify limitations in existing approaches—specifically regarding positional coherence, frequency utilization, and compatibility with pre-trained LLM priors. To address these, they propose two variants: Multi-Head RoPE (MHRoPE) and MRoPE-Interleave (MRoPE-I), utilizing a "spatial-reset" mechanism and improved frequency allocation strategies. The methods are evaluated across a wide range of benchmarks covering image understanding, video understanding, and visual grounding.

The reviewers appreciated the paper's systematic analysis of multimodal RoPE but raised significant concerns regarding the generalizability and experimental scope of the work. A primary critique was the exclusive reliance on the Qwen2.5-VL architecture for evaluation; they questioned whether the proposed guidelines and performance gains would transfer to structurally distinct VLMs or different parameter scales. They further noted that using Qwen-specific components might introduce confounding factors favoring the proposed method.

Additionally, reviewers questioned the method's robustness in extreme multimodal scenarios. Reviewers requested evidence of performance on multi-image interleaved sequences and long-context video extrapolation to verify that the "spatial-reset" and frequency allocation strategies effectively mitigate attention sinks without degrading long-term dependencies. Finally, there were concerns regarding technical novelty relative to the original MRoPE and requests for missing implementation details and theoretical justifications.

**Reviewer Concerns:**

Concerns addressed by Rebuttal (or at least partially addressed):

- Limited Model Diversity: A primary concern was that experiments were limited to the Qwen2.5-VL family. In the rebuttal, the authors conducted extensive new experiments on the structurally distinct Qwen3-VL architecture (4B and 8B variants). The proposed methods maintained their superiority, effectively demonstrating generalization across architectures.

- Robustness in Complex Scenarios: Reviewers asked for evaluations on multi-image sequences and long-context scenarios. The authors added results on the BLINK benchmark (interleaved images) and extended video context evaluation to 256K frames, demonstrating that their "spatial-reset" prevents attention sinks and handles complex inputs better than baselines.

- Implementation Details & Reproducibility: The authors updated the manuscript to include missing training details (optimizer settings, data composition) and clarified that the vision encoder was frozen to isolate the impact of the LLM's positional encoding, addressing concerns about confounding factors.

- Theoretical vs. Empirical Justification: The authors pointed to mathematical derivations in the appendix regarding frequency allocation and decay, strengthening the theoretical grounding of their interleaved strategy.

Outstanding concerns:

- Incremental Novelty: While the reviewer noted the method is a modification of MRoPE, I think the authors successfully argued that this is a "systematic fix" rather than a minor tweak. The identification of failure modes in existing specialized RoPEs and offering a unified solution constitutes significant practical value, even if the architectural change is subtle.

**Reviewer Scores:**

- Reviewer KSo7 (Initial: 4): likely to increase to 5. The reviewer’s main weakness was "Limited experimented VLMs." The addition of Qwen3-VL results partialy addresses this. The authors' response on the novelty concern also make sense.

- Reviewer PFUK (Initial: 4): likely to increase to 5. The reviewer requested better reproducibility details and broader experiments, both of which were provided.

- Reviewer T4pq (Initial: 6): likely to remain 6 or increase to 7. The reviewer was already positive, and the rebuttal clarified the compatibility with extrapolation methods, reinforcing the rating.

- Reviewer kJVi (Initial: 6): likely to remain 6 or increase to 7. The reviewer’s concern about single-instance validation was resolved by the Qwen3-VL experiments.

---

### Decision · Program_Chairs · 2026-01-26

Accept (Poster)